# Interdisciplinary Water Development in the Peruvian Highlands: The Case for Including the Coproduction of Knowledge in Socio-Hydrology



**Jasper Oshun** [1,*] **, Kristina Keating** [2] **, Margaret Lang** [3] **and Yojana Miraya Oscco** [4]

1   Department of Geology, Humboldt State University, Arcata, CA 95521, USA
2   Department of Earth and Environmental Sciences, Rutgers University, Newark, NJ 07102, USA; kmkeat@rutgers.edu
3   Department of Environmental Resources Engineering, Humboldt State University, Arcata, CA 95521, USA; margaret.lang@humboldt.edu
4   Department of Political Science, University of Toronto, Toronto, ON M5S3G3, Canada; yojana.mirayaoscco@mail.utoronto.ca
*   Correspondence: oshun@humboldt.edu

**Abstract:** Agrarian communities in the Peruvian Andes depend on local water resources that are threatened by both a changing climate and changes in the socio-politics of water allocation. A community's local autonomy over water resources and its capacity to plan for a sustainable and secure water future depends, in part, on integrated local environmental knowledge (ILEK), which leverages and blends traditional and western scientific approaches to knowledge production. Over the course of a two-year collaborative water development project with the agrarian district of Zurite, we designed and implemented an applied model of socio-hydrology focused on the coproduction of knowledge among scientists, local knowledge-holders and students. Our approach leveraged knowledge across academic disciplines and cultures, trained students to be valued producers of knowledge, and, most importantly, integrated the needs and concerns of the community. The result is a community-based ILEK that informs sustainable land and water management and has the potential to increase local autonomy over water resources. Furthermore, the direct link between interdisciplinary water science and community benefits empowered students to pursue careers in water development. The long-term benefits of our approach support the inclusion of knowledge coproduction among scholars, students and, in particular, community members, in applied studies of socio-hydrology.

**Keywords:** socio-hydrology; knowledge coproduction; integrated local environmental knowledge; education and training; community-based water development

## 1. Introduction

In the high-elevation Andes of Perú, agrarian communities, many of which are Indigenous, depend on local water resources for sustenance. Sustainable and lasting water management in the Peruvian Andes requires a complete perspective of water resources that incorporates physical science, social context, and the production of knowledge. Although human impacts often are not fully quantified in watershed studies, there is a growing need to integrate the combined effects of humans and hydrology to focus on "land-change science" [1,2] that integrates society with nature by way of socio-hydrology [3] or analyses of "waterscapes" shaped by both hydrological and social flows [4,5]. Here, we present a framework for explicitly including the coproduction of knowledge from scientific experts, students, and local knowledge-holders in the rural Andean village of Zurite and argue that such a framework is critical for sustainable water development.

Water resources in the Peruvian Andes are impacted by humans at global, regional, and local scales. Global-scale climate forcing accelerates the melting of glaciers, impacting

water supply (e.g., see [6,7]). Local and regional development projects, such as hydroelectric dams or mining operations, impact water use and drive laws affecting water accessibility (e.g., see [8,9]). Additionally, traditional Indigenous practices, dating to pre-Inca times, govern land management and aquifer recharge, and inform local water use and community water allocations (e.g., see [10]). These activities result in complex spatiotemporal effects on the water supply and create challenges for the study of water resources and the implementation of sustainable water management in the region. Carey et al. developed a socio-hydrology framework bridging natural and social sciences to model how downstream communities may adapt to changes in local water supply [11]. The focus of their study was the Santa River basin, a watershed in which communities, mining operations and hydroelectric dams compete for shrinking supplies of glacial meltwater. In addition to scientific data to understand water resources, Carey et al. incorporated five human variables—politics and economics, laws and institutions, technology, land and resource use, and societal response—into their framework. The framework provides a useful link between science and society, as these factors strongly influence communal water use and sustainable water management.

Here, we argue that, in order for socio-hydrology to not only link natural and social science but also to result in community-level adaptation to climate change, we must include a sixth factor: the collaborative coproduction of knowledge. Our approach to knowledge coproduction leverages community needs and knowledge with our scientific expertise, and trains a collaborative, interdisciplinary, and multinational cohort of students in community-minded approaches to water resources research. Including knowledge coproduction in the socio-hydrology framework, particularly when using an interdisciplinary, applied, and community-minded approach, provides a platform for knowledge transfer and mutual learning between all stakeholders in the project, and is critical for achieving community-level resiliency to climate change such as local sustainable land and water management.

## 2. Combined Socio-Hydrology Hydro-Social and Integrated Local Environmental Knowledge Conceptual Model

We used an integrated local environmental knowledge (ILEK) approach that focuses on the coproduction of knowledge to inform decision-making and action in Zurite (Figure 1). The ILEK approach was developed by Sato et al. [12–14] and is purported to increase a community's resiliency and ability to adopt environmentally sustainable changes. We used this approach to outline the knowledge actors who collectively produce the knowledge and inform sustainable environmental decision-making. All the knowledge actors in the ILEK approach contribute to three interconnected outcomes: knowledge coproduction, decision-making and action, and institutional change. In our framework, the knowledge actors were the local knowledge holders, i.e., the local stakeholders from the community of Zurite, the scientific experts (geologists, geophysicists, engineers, hydrologists, sociologists), and the U.S. and Peruvian students.

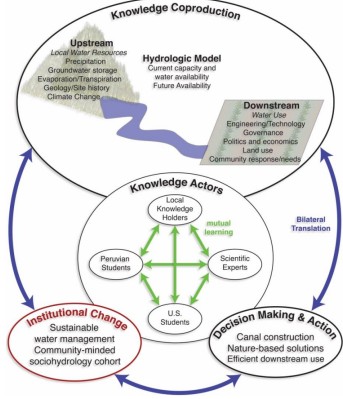

**Figure 1.** The conceptual model of knowledge coproduction, applied to a hydro-social framework [11].

scientific experts, local knowledge-holders, and U.S. and Peruvian students combine their forces, to learn from one another and coproduce upstream and downstream knowledge. The result is local environmental knowledge (ILEK) [12–14] that integrates upstream and downstream disciplines, and western and traditional science, to inform decision-making and promote institutional change, such as sustainable water management.

The knowledge coproduced by this set of actors applies the hydro-social framework [11], including "upstream" and "downstream" knowledge, to develop an ILEK linking upstream water resources to downstream water use and decision-making that, in turn, guides land use/management, resulting in potential changes to upstream water resources. Here, upstream knowledge focused on water resources originating in the Upper Ramuschaka Watershed (URW), the primary water source for the village of Zurite, and included the disciplines of hydrology, geology, geomorphology, and geophysics. Downstream knowledge focused on water use, local knowledge, and decision-making in Zurite, and includes the disciplines of engineering, sociology, history, and community studies, with a focus on canal reconstruction and the potential for adaptations to water use. Two major goals of the project were sustainable water management within the village of Zurite and developing a trained cohort of socio-hydrologists. By including knowledge coproduction and the training of an interdisciplinary and multinational cohort of students, the ILEK produced will benefit from the diversity of knowledge actors (in terms of age, subject area expertise, ethnic and cultural background, etc.) who, together, will be better at solving complex environmental problems [15,16]. All knowledge actors will benefit from a platform facilitating knowledge exchange and mutual learning. Additionally, students will benefit when trained to understand the complexities of upstream and downstream disciplines affecting sustainable water management and are empowered through their contributions to the project's success.

We applied this combined hydro-social/ILEK framework to a two-year water development project that took place in Zurite (population 3600), Peru, named Bonanza en los Andes (referred to as Bonanza). The structure of this paper closely follows the three interconnected themes of our project: science, society, and knowledge coproduction. First, we summarize the upstream knowledge relating to water resources in the Vilcanota Watershed, which includes Zurite, and the URW. Next, we summarize the downstream knowledge and discuss water and society in the Peruvian highlands. Third, we describe our approach to knowledge coproduction, which blends upstream and downstream disciplines and focuses on training cohorts of students who contribute to an ILEK. Finally, we report the knowledge produced, as well as the short-term and anticipated long-term impacts of our project on the community of Zurite and on participating students.

## 3. Upstream and Downstream Knowledge: Water Resources and Use in Andean Perú

### 3.1. Upstream Water Resources throughout the Vilcanota Watershed

Perú holds more freshwater resources per capita than any other South American country; however, the longitudinal barrier of the Andes forms a barrier between the moist eastern flanks, draining to the Amazon Basin, and the rain shadow desert along the populous Pacific coast [17]. High elevation peaks trap moisture and high-elevation, headwater streams coalesce to form large eastward-flowing rivers, such as the Vilcanota-Urubamba (Vilcanota) River.

The Vilcanota River originates in the Altiplano near Ausangate (6372 m.a.s.l.) and flows northwest through the Sacred Valley, north of Cusco (Figure 2). The Vilcanota Watershed totals 11048 km$^2$ at the confluence of the Urubamba and Lucumayo Rivers in Santa Maria (elevation 1200 m.a.s.l., [8]), and is the second most glaciated tropical watershed in the world. However, by 2016, only 1.28% of the watershed was glaciated, reflecting a 30% reduction since 1985 [18–21]. The high elevation portions of the watershed, including the village of Zurite, have a present-day Köppen–Geiger climate classification of CwB [22].

The hydrograph of the Vilcanota River is driven primarily by precipitation, which is concentrated between November and April (less than 4% of annual precipitation falls between May and August) [8]. Glacier meltwater contributions to the Vilcanota River are estimated to be less than 3% annually [23]. The vast majority of the Vilcanota Watershed (~78%) is part of the puna biome: a seasonally dry grass and shrubland environment existing above the tree line and below the permanent snow line, along the spine of the Andes from central Perú to central Chile and Argentina [24–27]. Groundwater contributions from the puna sustain perennial headwater streams and account for nearly all the annual discharge in the Vilcanota River [8,23]. The city of Cusco, the regional capital and home to over 420,000 people [28], draws approximately 90% of its water from the combined sources of Laguna Piuray—a lake fed by an approximately 26 km² catchment area draining the puna—and the Vilcanota River, which is primarily groundwater-fed [29]. Within the Department of Cusco, over 20% of its inhabitants are without access to a permanent water supply [30]. The percentage without access to safe water resources rises to 42% in the rural province of Anta (55,000 inhabitants), of which Zurite is one of 9 districts [31].

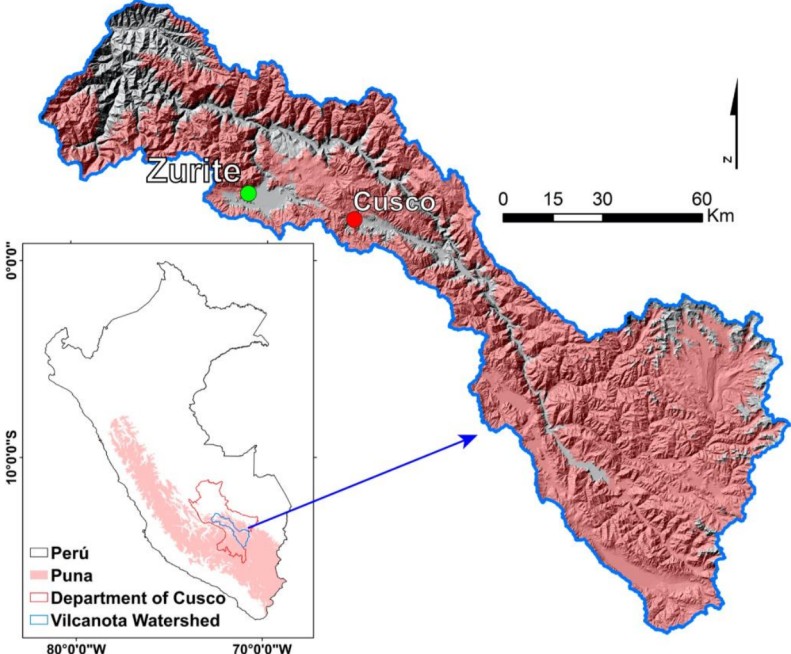

**Figure 2.** Zurite is located approximately 40 km to the northwest of Cusco, within the 11,048 km² Vilcanota Watershed [8]. The Vilcanota flows to the NW and is entirely within the Department of Cusco, outlined in red on the inset map. Red shading indicates the extent of the puna biome in Perú and within the Vilcanota Watershed [25]. The Sacred Valley is the gray region (lower elevation than the puna) directly north of Cusco and Zurite.

Much of the puna exists in landscapes that are sculpted by glaciers with distinctly glacial morphology, such as steep cirque walls, hanging valleys, moraine deposits, and, importantly, gently sloping valley floors in which peat-forming, seasonally saturated wetlands—known as bofedales—are often found [24,26,32]. Bofedales act as shallow aquifers, filling in the wet season, due primarily to groundwater contributions from up-slope, and draining to streams throughout the dry season [33]. As temperatures warm and glaciers melt, the puna biome is predicted to expand to higher elevations [34], suggesting even greater proportions of puna-derived runoff from these watersheds. Seasonal variability in precipitation in the Vilcanota Watershed is projected to increase [35], which, importantly, could lead to a delay in the onset of precipitation in September through December, a time critical for planting the most widely grown crop in the Department of

Cusco, corn [36]. The effect of changes in the timing and quantity of precipitation and the impacts on water resources, particularly in the dry season, are poorly understood.

*3.2. Downstream Knowledge: Water and Society in Andean Peru*

3.2.1. Water Rights in Perú

To understand the societal impacts on water use in Zurite, it is helpful to understand the historical context of water rights in Perú. Both the General Water Law of 1969 and agrarian reform in the 1970s had a positive impact on the ability of local communities to regulate their water use. These changes placed water management decisions in the hands of local entities, known as Comisiónes de Regantes (Water Users' Commissions), and redistributed land from large haciendas to indigenous families (see, e.g., [37]). The decentralization of power allowed for increased local autonomy over water and land management (see, e.g., [31]). However, in the 1990s, a wave of neoliberal economic policies resulted in expanded mining operations and hydroelectric dams, which increased water demand [38]. To support these operations, decision-making power was concentrated in the hands of companies, despite their small numbers compared to stakeholders within the watersheds [39,40]. More recently, there has been additional pressure on the limited water supplies, due to an increase in the amount of water exported, carried in the form of vegetables [6]. The increased demand for water has disproportionately affected small, rural, and predominantly indigenous communities and, in some cases, led to conflict over water resources between mining companies and local stakeholders (see, e.g., [5,6,41]).

Over the final decades of the 20th century, water resources in Andean Perú were targeted for privatization by the World Bank, to (1) decentralize decision making, (2) create and protect private property rights for water, and (3) establish a market for water to improve water use efficiency [37,42]. In actuality, aging infrastructure and flood irrigation practices result in poor water-use efficiency and 86% of national freshwater use by the agricultural sector [17] and market forces resulting from the privatization of water are not likely to incentivize water conservation [37]. Alternative approaches for water management, in which decision-making power lies with the community, have been shown to increase efficiency and protect Indigenous communities' access to water by leveraging traditional water management practices (see, e.g., [37]). As presented in our conceptual model above (Figure 1), we argue that decision-making is best informed by the coproduction of knowledge from a diverse body, including scientific experts, local knowledge-holders, and students.

3.2.2. Indigenous Andean Communities and Allin Kawsay

A core principle of Allin Kawsay, the Indigenous Andean cosmovision, is to live in harmony with the environment and the community [43,44]; this principle guides the communal laws and customs of water allocation within Indigenous Andean communities [45]. Allin Kawsay exemplifies traditional ecological knowledge (TEK) and refers to the principles of "good living" in Quechua. Similar to the TEK of the Indigenous peoples of North America, in Allin Kawsay, all aspects of a physical space—mountains and other landforms, plants, animals—are interconnected and deserving of the same respect and treatment [46]. Furthermore, the health of the mountains, other landforms, plants and animals serve as a reflection of the health of the community. [47,48]. Mountains, in particular large peaks, are homes to powerful spirits, or apus (see, e.g., [45,49]). The apus provide water, but, when dissatisfied, can also cause lake outburst floods that are interpreted as ominous signs [50]. Allin Kawsay forms a holistic world view that emphasizes a collaborative relationship between humans and the physical space, rather than an extractive one.

Although Indigenous Andean communities have been resilient over generations and have historically adapted to changes in climate, in part due to the guidance of Allin Kawsay, the current accelerated rates of climate change, combined with a lack of government resources, endangers access to local water supplies, resulting in local hardship [51,52]. Many Indigenous communities are reliant on agricultural production for subsistence and

possess limited alternatives for producing a living, yet they have typically been excluded from water resources allocation discussions (see, e.g., [8,53]). A 2009 law, which defined water as national property, sought to prevent the privatization of water and increase water use efficiency [54]. This law created a national administration for the use of water, as well as providing agency to local governments to manage local water resources [39]. Nevertheless, a frequent turnover in government leadership, particularly at the national level, may result in changes with detrimental effects on Indigenous communities that are best countered by longer-term planning (exceeding 10 years) within these communities [21].

Much of the past research has focused on examining the impact of melting glaciers on Indigenous communities' access to water (see, e.g., [8,11,41,52,53]). However, Indigenous Andean communities that are dependent on non-glacial water sources, such as the puna, are also vulnerable to changes in climate and/or policies that affect access to local water resources. To develop sustainable water management within the Indigenous community of Zurite, which depends on water originating from the puna for sustenance, we used an interdisciplinary approach to knowledge coproduction that integrates the perspective of Allin Kawsay with community needs, upstream and downstream disciplines, and student and local training to produce ILEK [13,55]. In addition to building local capacity and community resiliency within Zurite, this approach will train a culturally educated cohort of socio-hydrologists and provide a framework that can be adapted to other communities. Our approach focuses on issue-driven and solution-oriented science, and the critical role of the local community in transforming the relationship between the environment and society, to ultimately achieve sustainable futures [13].

## 4. Framework for the Coproduction of Knowledge

### 4.1. Project Goals and Approach to Knowledge Coproduction

We designed a year-long education program to support the project's objectives of water resources research spanning upstream water sources in the URW and downstream water uses in the community of Zurite. We leveraged Dr. Oshun's longstanding relationship with Zurite (he lived and volunteered in Zurite in 2003 and has since returned to support several development projects), such that we began our project with a collaborative spirit, uniting U.S.-based scientists and students with Peruvian students and the leadership in Zurite. The municipal government, Farmer's Union, and Water Users' Commission expressed a need for canals, and we began our project with an agreement to collaborate on the design and construction of irrigation canals, which we incorporated into our upstream and downstream water resources research. Both the research and the irrigation canal collaboration would contribute to ILEK via a continuous exchange of ideas and knowledge coproduction among the diverse group of participants. A guiding principle of our approach was that all knowledge-holders (see Figure 1; local knowledge holders, U.S. students, Peruvian students, and scientific experts) are valuable assets and are thus critical to the overall project success.

We designed the curriculum of our project to guide the students through the application of research, to the benefit of the community of Zurite. The curriculum included topics in the upstream disciplines of hydrology, geology, geophysics, and ecology, and the downstream disciplines of engineering, sociology, history, and community studies. We leveraged our experiential and academic expertise to include TEK and Allin Kawsay within the curriculum, to provide context and a deeper and more holistic understanding of water resources, usage and the needs of the community.

As longer-term student participation in projects has been shown to maximize the positive and lasting impact on students [56,57], our education program was designed to span a full year, with three components each year: a preparatory course in the spring, an international immersive field experience in the summer, and an independent research project in the fall (summarized in Figure 3). U.S. students from the primary research institute (Humboldt State University, HSU) committed to the program for at least half the year, with the majority of students participating in all three components. Peruvian students

and U.S. students who were not from HSU participated in the summer international research experience and, in some cases, also participated in the fall independent research. The project ran for two years (2018 and 2019). The students' backgrounds and each component of the program are summarized below.

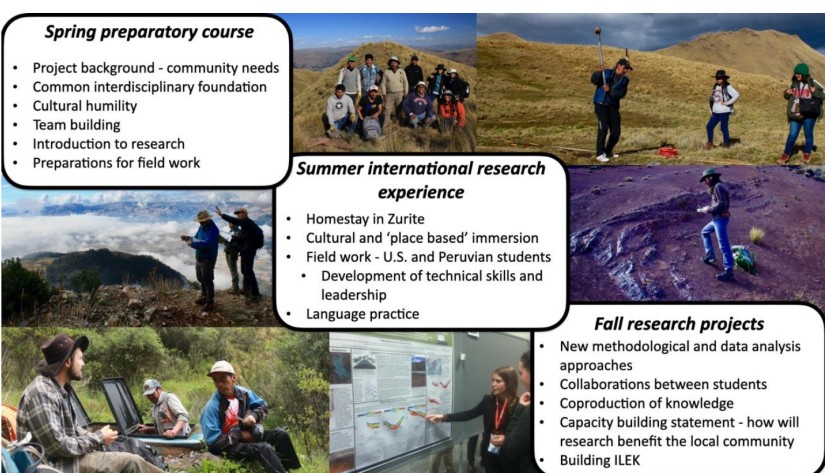

**Figure 3.** The educational framework for our Bonanza project. We embedded the summer international research experience within coursework to maximize the impacts and foster sustained learning.

### 4.2. Student Participants

Over two years, 29 students participated in either the coursework, the international field experience, or both (12 students in 2018 and 17 students in 2019). Students came from undergraduate (18) and graduate (11) programs in the United States (23) and Perú (6). Students were pursuing degrees in geology (13), environmental resources engineering (7), geophysics (2), hydrology (2), physics (1), geography (1), film (2), and environment and community (1). Two students participated in both years of the program. The 22 HSU students took 1–4 semesters of preparatory and research-focused course work, spanning the international field experience.

### 4.3. Spring Preparatory Course

We designed the spring preparatory course to build a common foundation of knowledge among students with diverse educational backgrounds, and to encourage students to apply their disciplinary skills to pursue individual research. We built foundational knowledge through introductory lectures focused on the disciplines necessary to achieve our objectives of water resources research in the URW and the construction of irrigation canals in Zurite: regional geology, the puna landscape, the Andean cosmovision, Incan and contemporary irrigation practices, and the community of Zurite. These lectures were accompanied by exercises and discussions that combined to form an interdisciplinary focus on water, which included physical hydrology, agricultural water needs, regional and local water management, and the issues of water quantity and quality encountered by local communities and community organizations. We fostered feelings of empathy and connectedness, developed communication skills, and engaged students in teamwork through exercises connecting our scientific objectives to the points of view and needs of the Indigenous and farming community of Zurite. These exercises were designed to instill a sense of respect for people of other cultures, which, through the international summer research, grew into intercultural competence or cultural humility (see, e.g., [58–60]).

Concurrently, we guided students through the research process by leveraging their disciplinary and newly acquired skill sets. Students worked independently to develop research questions, conduct literature reviews, generate hypotheses, and construct a plan for field research. The full list of projects and skills gained by the students associated with each project are provided in Table 1. Example projects included a simple hydrologic model

to inform site selection for the installation of stream gages, a flight plan to collect drone data to generate a digital elevation model, an assessment of current and projected irrigation needs, a calculation of crop-specific irrigation needs and irrigation canal capacities in Zurite, and the hydraulic engineering of proposed sections of irrigation canals. These projects integrated our growing data set with the varied skillsets of the students in the class, and directly contributed to our objectives of understanding water resources in the Andean puna and informing sustainable water use in the community of Zurite. At the conclusion of the semester, students presented a progress report and a testable hypothesis, and identified their responsibilities and needs, to test their hypotheses in the summer field research.

**Table 1.** List of student projects, contributions to ILEK in Zurite, and the skillsets acquired by students. For the locations, U indicates upstream, and D indicates downstream.

| Project | Location | Contributions to ILEK | Skill Sets |
|---------|----------|----------------------|------------|
| Geologic field mapping | U | Geologic map of URW, including identification of potentially active faults and landslide hazards | 4-dimensional thinking, integration of Spanish language geology terms, cross-cultural collaborations in the field |
| Digitizing geologic map | U | Spatially oriented geologic map | GIS skills, interpreting field notes, 4-dimensional thinking |
| Drone flight plan and data collection | U | 1-m digital elevation model of URW, videos of groups working in the field and of the landscape | UAV pilot license, flight experience at high elevation, structure from motion data analyses |
| Geophysical Analysis | U | Application of geophysical methods to determine the subsurface structure and inform aquifer storage and hydrologic flow pathways | Field survey design, team management, data processing analysis, and interpretation, AGU poster presentations |
| Slope stability analysis | U | Identification of landslide hazards above Zurite | Application of published model, geospatial skills, interpretation of model sensitivity to model parameters |
| Installation of hydrologic monitoring equipment | U | Continuous recording of precipitation and temperature, discharge distributed throughout the URW | Identification of suitable sites, rating curve construction, cross-cultural learning, data analysis and quality control, AGU presentation |
| Installation of deep monitoring wells | U | Continuous measurements of groundwater resources beneath hillslope and in bofedal (2019–2021) | Contract work, interdisciplinary learning, cross-cultural learning, language practice |

**Table 1.** *Cont.*

| Project | Location | Contributions to ILEK | Skill Sets |
|---|---|---|---|
| Distributed discharge measurements | U | Spatiotemporally distributed discharge at 1–3-month intervals (2019–2021) to identify connections between landscape structure and hydrologic productivity | Interdisciplinary learning, field skills and technology to apply to senior theses, contract-based employment, cross-cultural learning, language practice |
| Estimates of seasonally dynamic water storage in URW | U | Connection of geology and puna landscape structure, including geophysics, to rainfall-runoff metrics and total water yield | Application of mathematical analysis presented in recent literature to URW stream data, processing data in R, experience teaching other students |
| Soil characteristics and plant water status in the URW | U | Characterization of the URW within the puna biome, soil classification, plant water availability and source water identification | Application of research methods to a new environment, opportunity to teach others in the classroom and field |
| Distributed evapotranspiration model Zurite | U | Development of model to predict ET | Geologic field mapping, ground-truthing of a remotely sensed model, cross-cultural learning, language practice |
| Hydrologic modeling using MODFLOW | U | Identification of wet, low gradient regions (bofedales) and springs, guided future hydrologic field measurements | MODFLOW, model sensitivity analyses, teamwork through the integration of datasets (geologic, hydrologic, seismic) through collaborations with student colleagues |
| Quantification of water storage in bofedales | U | Hydraulic properties of bofedales, estimation of dynamic storage and contributions to streamflow | Collection of data in the field over two seasons, interpretation of multiple methods, spatial analyses, and the connection of results to broader project and community needs |
| Stage-discharge rating curve for diversion weir at outlet of URW | D | Relationship to quantify continuous discharge from URW and connect to water demand downstream | Application of engineering skills for community benefit, analysis of existing data, connection to necessary field measurements |

**Table 1.** *Cont.*

| Project | Location | Contributions to ILEK | Skill Sets |
|---|---|---|---|
| Irrigation needs: current and under future climate scenarios | D | Quantification of total irrigation water demand, projections of changes to supply (small) and demand (large increase under a warming climate water use and identification of opportunities to boost local resiliency | Application of engineering skills to the benefit of the local community, cultural humility, interpretation of risk to community, and potential resiliency |
| Hydraulic modeling of existing and proposed canal network HEC-RAS, and estimated cost | D | Design of proposed canal, including material and labor cost estimates | Application of engineering skills for community benefit, cross-cultural learning, and cultural humility |
| Water Quality in Zurite | D | Distributed tests in new potable water system showed good water quality in 2018 | Application of engineering coursework, communication with the community, cultural humility |
| Film—interviews of project participants and community | D | Zuriteños empowered to voice their concerns to outside scientists, opportunity to learn across the community and from the Bonanza group | Constructing narrative arc, developing questions, Spanish Quechua language, data management, empathy, cultural humility |

### 4.4. Summer International Research Experience

We led month-long research and community development trips to Zurite in 2018 and 2019. Students stayed in the homes of three different host families, with shared meals and discussions at the home of a centrally located host family. The homestays and engagement with the community encouraged open-mindedness, and exposed students to a diversity of values and cultural backgrounds e.g., [61].

Much of the logistics and the execution of fieldwork were led by the students, with the program faculty providing guidance in the field and through evening discussions. In the URW, fieldwork included geologic field mapping, the installation of hydrologic monitoring equipment—including rain gages, subsurface moisture probes, and groundwater monitoring wells—soil substrate measurements, and intensive geophysical surveys. In Zurite, fieldwork included surveys of the irrigation water distribution infrastructure, measurements used to design the construction of new canals, and gathering of water use information from local water officials. U.S. students worked alongside Peruvian students, providing mutual learning opportunities for field techniques such as geologic mapping, hydrologic monitoring, geophysical surveys, drone flights, rotary drilling, and well installation, as well as sharing customs. The unique experience of working in Zurite, including the strong tropical sun, the exhilarating thin air, and unique smells, tastes and sounds, provided a learning experience that cannot be replicated in a classroom [62].

Members of the community contributed to our fieldwork as part of the organized faena, or work patronage paid to the community. We offered training focused on gaging rivers, learned local perspectives, and explained our scientific objectives. Incorporating local representatives in our fieldwork allowed for an exchange of ideas between the local knowledge-holders and the students, and also demonstrated the community's investment in our research and in the water development project. The shared experience between

students and volunteers provided an opportunity to both gain linguistic and cultural capital (see, e.g., [59,63]) and carry forward an experience of seeing the world through someone else's eyes.

Our role in Zurite, beyond the scientific objectives in the URW, and our contributions to the design and construction of irrigation canals, was as facilitators of knowledge coproduction contributing to ILEK. Our approach has been inclusive and participatory. Rather than communicating in one direction from science to citizens, we followed the best practices of including stakeholders (see, e.g., [64]) to identify community needs to help inform best practices in society-environment interactions (see, e.g., [65]).

*4.5. Fall Research Projects*

In the fall, we advised independent student research, and coordinated discussions to encourage collaboration between students in weekly meetings. Embedding the students' international research experience within the coursework, and participation in the follow-up semester, instills lasting learning [56]. We introduced new methodological approaches through four introductory labs, in which students explored and processed drone-derived imagery, precipitation and runoff data to construct a water balance, seismic data to determine water-holding units in the subsurface, and canal hydraulics data to design irrigation canals. As the semester progressed, we loosened the structure, allowing more time for students to creatively explore their individual research topics. We met with students to review their progress, in a series of checkpoints designed to lead students through analyses, interpretation, and preparation for oral and written reports. Importantly, students were asked to include a "capacity-building statement", outlining a plan to integrate the knowledge they produced into the community sphere. By explicitly including the needs of the community, we guided students to apply their research, fostering a deeper appreciation for the host country and, importantly, an enhanced understanding of one's agency and responsibility as a local and global citizen (see, e.g., [66]). Table 1 presents a list of student-led projects, either in the classroom, in the field, or both, each project's contribution to ILEK, and the skillsets acquired by the students.

## 5. Knowledge Coproduced and Project Impacts

*5.1. Upstream Knowledge: Upper Ramuschaka Watershed and Local Water Resources*

The primary water source used for irrigation in Zurite is the 2.12 km$^2$ Upper Ramuschaka Watershed (Figure 4). The URW presents as a typical puna landscape showing glacial morphology, including steep headwalls of bedrock or grasslands and low gradient bofedales. The underlying geology is composed primarily of sandstones, conglomerates, quartzites and mudstones, with isolated carbonates in the western headwaters of the URW. The sedimentary units are part of the San Jeronimo Group, which is Eocene to Oligocene in age [67]. The sedimentary units typically dip to the southwest; however, there are a number of smaller-scale folds and faults in the URW. There is a large quartz monzodiorite intrusive complex, which is likely part of the Yauri-Andahuaylas Batholith (42–30 Ma) along the western boundary of the URW [67–69].

We measured annual precipitation in WY 2019 to be 752 mm, and in WY 2020 to be 825 mm (details found in [70]), which is similar to the 38-year average of 855 mm in Anta (10 km away, [71]). Over our study period, approximately 50% of precipitation occurred between December and February, and only 1–3% between June and August. Runoff accounted for 62–80% of annual precipitation and was also highly seasonal, with only 10–18% occurring in the dry season. Importantly, however, streamflow measured at the diversion weir located at the outlet of the URW never fell below 11 L/s. Streamflow in the URW is fed directly by groundwater draining from hillslopes or the slow drainage of bofedales, which hold large volumes of water and cover 11.5% of the URW. We estimate, conservatively, that seasonal storage in bofedales amounts to about one-half of dry season runoff in the URW. The presence of bofedales in the source watershed of Zurite, and their

capacity to seasonally store and release groundwater to sustain perennial flow, represents a built-in source of natural resiliency and is the focus of ongoing research [70].

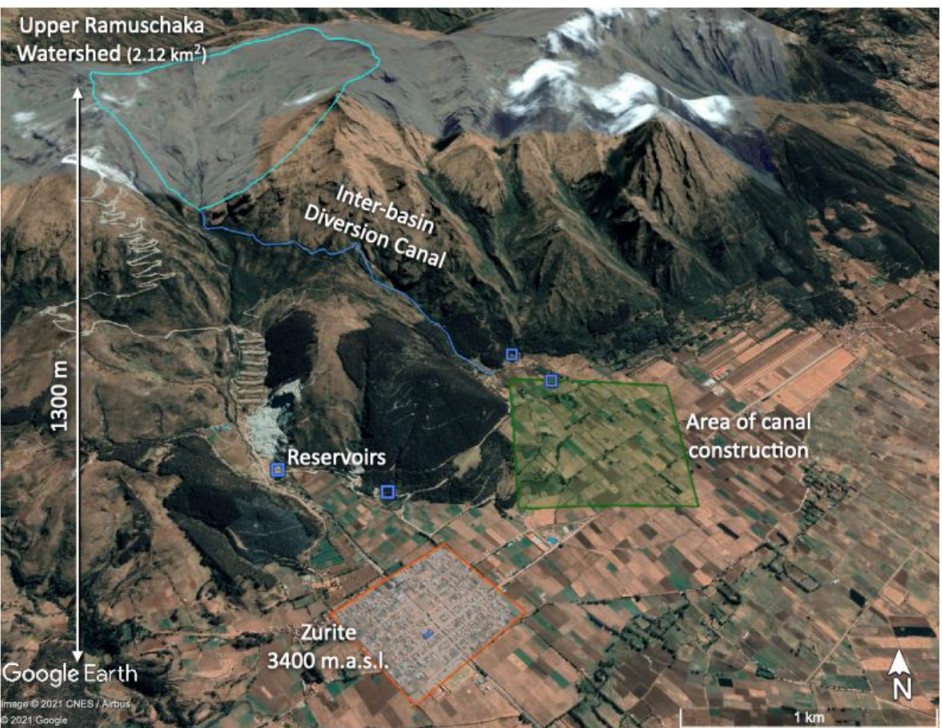

**Figure 4.** Zurite and the Upper Ramuschaka Watershed (URW). The URW is located 1300 m above the village, within the puna biome. Water is often diverted from the URW via an inter-basin canal for irrigation to the east of Zurite. Blue squares show the locations of 4 surface reservoirs, and the green-shaded area shows the area in which we collaborated with the local community to construct 1.3 km of irrigation canals (map and photo below).

*5.2. Downstream Knowledge: Water Use and Governance in Zurite*

Through discussions with local leaders and water users, we produced the following summary of how water is used in Zurite within the context of water resources derived from the URW. Systems in Zurite can be described as 'Indigenous', meaning that irrigation is for the most part carried out in a manner similar to traditional Andean practices (see, e.g., [72–74]). Water draining from the URW is diverted via a concrete weir and inter-basin transfer canal, where it is routed through a system of canals to flood and irrigate crops to the east of Zurite (Figure 4). Corn, grown in the wet season, represents approximately 80% of the agricultural production across Zurite's 1200 ha of cropland, with lesser production of potatoes, quinoa, wheat, fava beans, and forage for animals. The irrigation need is most acute over the dry winter, from May to September (primarily to grow fava beans) and, crucially, to irrigate fields recently sowed with corn before the onset of the rainy season, which begins in September (Figure 5).

Agricultural plots, known as chacras, are typically 0.25 to 1 ha and are owned and operated by individual families. Because the Ramuschaka Watershed is within the District of Zurite, the community of Zurite has the right to form and elect a local Water Users' Commission. Similar to other Andean communities, the Water Users' Commission manages turnos, or turns, in which users pay 15 soles/hectare (~USD 5/hectare) for the right to divert water from the extensive canal network to flood-irrigate their chacras. An aging canal infrastructure system and the nature of flood irrigation result in poor water-use efficiency. The president of the commission is elected every 4 years and is responsible for regulating and ensuring equitable water allocation to users in the community. In order to operate, the Water Users' Commission in Zurite must follow the regulations outlined

by both the regional Water Administrative Authority in Cusco and the National Water Authority and collect and pay taxes to both administrative bodies. According to the Zuriteños, only rarely do these payments return to directly benefit the local community through irrigation system improvements. Instead of incorporating local best practices of land and water management, government institutions often seek to impose a homogeneous system of water infrastructure across diverse agrarian communities.

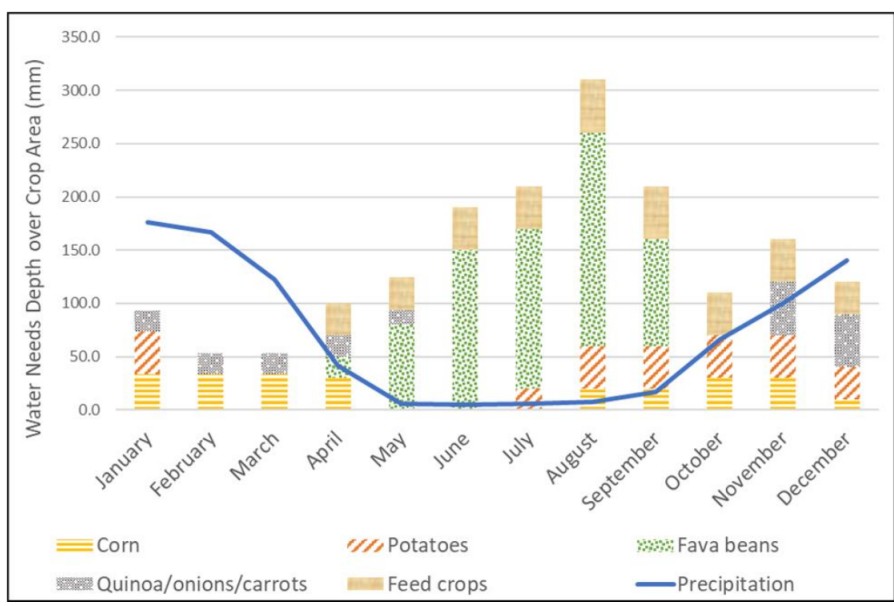

**Figure 5.** Average monthly precipitation (mm) and typical monthly irrigation needs (mm) for major crops grown in Zurite. The need for water is most acute in the dry winter to irrigate fields of fava beans, feed crops, and recently planted corn (August–November). Figure by W. Wunderlich and A. Virgil.

The efficient use of water resources in Zurite is challenged by limited storage, an aging infrastructure, and, in at least one case, a lack of community-based knowledge. The community operates five reservoirs with a total storage capacity of 10,500 m$^3$. This volume is substantially below dry-season irrigation demands and necessitates some fields being left fallow [75]. These reservoirs do not, nor were they intended to, function as long-term storage but instead operate as water elevation controls, allowing the multiple low flow-rate water sources to accumulate prior to delivery. The reservoirs are filled and drained approximately daily during the dry season, as water users schedule irrigation delivery on a rotating schedule to meet individual families' needs. Thus, the disruption of water delivered from both local and regional water supplies for even a short time can threaten a family's crops and livelihood. The construction of these systems across steep and, in some cases, unstable terrain makes irrigation disruptions extremely common. Figure 6a illustrates the construction variety throughout most of the system—concrete canals with control gates that minimize seepage water losses run parallel to and in series with hand-dug, earthen canals. Figure 6b shows a previously well-designed and controlled canal, constructed by the municipality, after it was damaged by a debris flow in March 2019. The debris flow was caused by a poorly designed and hydraulically mismanaged larger canal in the upper basin that was installed by regional authorities to transfer water from the outlet of the URW to an adjacent small drainage channel. Here, national water organizations proceeded without the involvement of local knowledge in the canal design, which led, in part, to this catastrophic debris flow and the destruction of the canal and surrounding crops of corn.

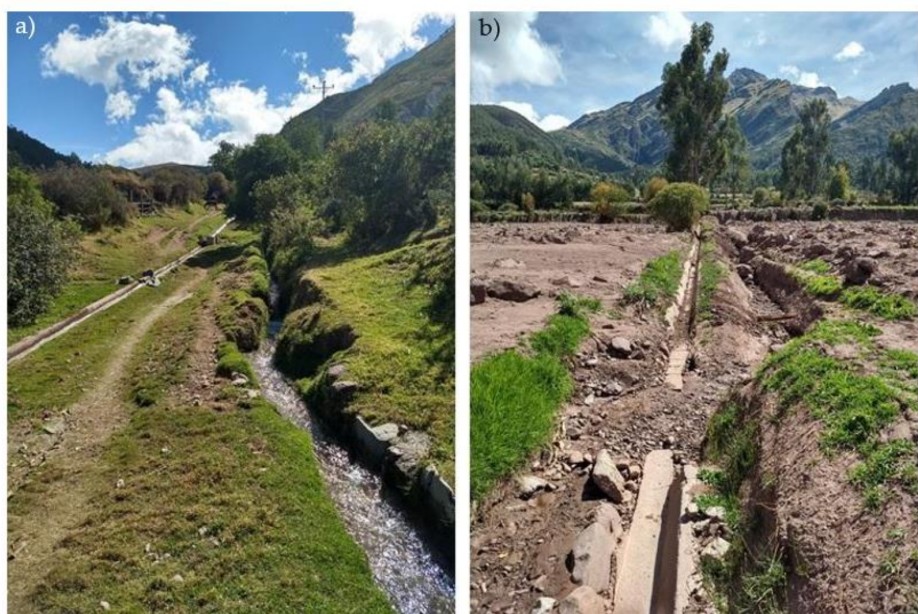

**Figure 6.** (**a**) A modern concrete canal with control gates adjacent to an earthen canal. (**b**) Destruction to concrete canal following March 2019 debris flow.

*5.3. Project Impacts on the Community of Zurite*

5.3.1. Immediate Infrastructure Benefit

The Bonanza project has resulted in immediate and longer-term benefits to the community of Zurite. Over the two years of the program, we collaborated with the municipal government, Farmer's Union, and Water Users' Commission in Zurite to negotiate, plan, and execute a USD 71,283 canal development project. Our monetary contribution to the project, funded by Geoscientists Without Borders, totaled USD 20,000. We advised student researchers, in collaboration with Zurite's engineers, in designing the canals. In March 2020, the community of Zurite finished building the 1.3 km of irrigation canals (Figure 7). These canals extended the irrigation system, to provide water to and boost the crop yield of land owned and farmed by over 100 families.

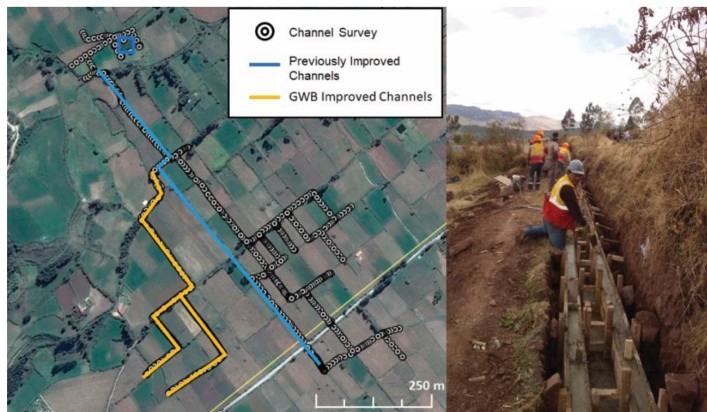

**Figure 7.** The left panel shows the area of canal improvements (green-shaded area of Figure 4). Circles show the extent of the irrigation canal network, with the blue square indicating the location of a surface reservoir that regulates flow to the canals. The blue line shows canals constructed by the community prior to 2018. The yellow line shows the extent of the 1.3 km of canals built through the Geoscientists Without Borders-funded Bonanza collaboration with the community of Zurite. The entire project totaled USD 71,283 and GWB's contribution was USD 20,000. The photograph on the right shows the early phases of construction in fall 2019.

### 5.3.2. Benefits of Knowledge Coproduction: Identification of the Risks and Opportunities of Current Water Resources and Irrigation Practices

Two examples illustrate our inclusive approach to knowledge coproduction, the learning that occurs in both directions, and the initial steps we have taken to develop an ILEK with the community. First, we learned of the principal concerns within the community through discussions with Zuriteños. An elder related, "We are worried about climate change and we want our children to as well . . . we think the educational aspect is the only way that we will have sustainable development in our communities." We agree that education and local training can directly contribute to the ILEK necessary for sustainable water management. As the elder expressed, younger generations are viewed as the inheritors of the land who, one day, will assume positions of authority and responsibility. Through our conversations with the community, we have gained a deeper knowledge of the issues at hand, and we are better positioned to work collaboratively to produce, share, and use knowledge [64].

We interpreted the elder's comment as a need for bottom-up approaches to building local capacity through training and the coproduction of knowledge to form an ILEK. Our response to this comment was to train locals in basic hydrological monitoring. Although in their early stages, these training initiatives have boosted the local capacity to monitor local water resources and inform water management.

Second, our fieldwork in the URW, one vertical kilometer above Zurite and a 50-minute car ride, or approximately a 4-h walk, provided an opportunity for local representatives to see their local water source. The remoteness of the URW disconnects Zurite's primary water source from the community. Tomás Ruiz López, the Community President of Zurite, remarked in 2019 that he had not seen the bofedales of the URW in twenty years. By working alongside our research team, Sr. Ruiz López and other representatives from the community have gained a new perspective and ILEK that will inform land management decisions in the community.

In order to disseminate our scientific findings on the importance of bofedales, and our recommendations for land conservation, we have written a progress report (2020), with a final report due in 2021, and we have presented our findings at two town hall meetings (2019 and 2020) and at a virtual symposium (2021). Our inclusion of community concerns in our reports, and our willingness to contribute our knowledge to ILEK in Zurite builds trust, and also leads to adaptive capacity and enhanced resiliency (see, e.g., [76,77]). The developing ILEK of our collaborative coproduction of knowledge identifies the risks of climate change-induced precipitation volatility, inefficient irrigation practices and limited water storage, frequent disruptions to infrastructure, and the opportunities of nature-based solutions (see, e.g., [10]), focusing on bofedal conservation or the introduction and expansion of more efficient irrigation practices, such as sprinkler irrigation.

We are currently working with the community to design and install up to ten sprinkler systems with flexible, replaceable, and relatively cheap parts. These sprinklers will connect to the existing canal network to increase water use efficiency, thus decreasing the total water demand on water resources from the URW building resiliency through adaptive water management. The presence of the sprinkler systems may also serve as a training model for neighboring communities and thus transfer resiliency beyond the community of Zurite.

### 5.4. Impacts on Student Participants

We used qualitative student commentaries to assess the benefits of knowledge coproduction on participating students. In particular, we focus on the high-impact practices of embedding our program in academic coursework [56], and increased self-confidence to pursue future research opportunities [57]. Below, we organize the impacts to students around three themes—the interdisciplinarity and applied nature of knowledge coproduction, the explicit inclusion of community knowledge and expressed needs, and the increased feelings of belonging and empowerment felt by students.

Bonanza leveraged the disciplinary knowledge of a diverse set of students and applied this knowledge to the objective of understanding water resources in the puna and building capacity and resilience in Zurite. Students cite the acquisitions of interdisciplinary perspectives as one of the primary benefits of research and study abroad [57]. A geology student, who participated in our program in 2018 wrote, "This project required me to utilize all of my past research and academic experience to work in a team environment with students, faculty and scientists from the U.S. and Perú to study the geology, geophysics and hydrology of a rural watershed in the Peruvian Andes." An environmental resources engineering (ERE) student from 2019 commented, "It was so valuable to get to work with multiple disciplines and gain new perspectives. Definitely life-changing." These sentiments were shared by a Peruvian student who participated from 2018 to 2020, "To be part of this research team was a great honor for me due to the learning that took place and, thanks to the members of this organization, I learned a great deal about geologic mapping, hydrogeology, hydrology, and geophysics."

Bonanza explicitly incorporated the knowledge and needs of the community of Zurite in the student research and learning experience. Whereas the experience of living and researching abroad is a high-impact practice that leads to greater student engagement [78,79] and builds a community that boosts student success (see, e.g., [80–82]), our program trained students not only to apply their knowledge for the betterment of the community but also to identify specific adaptations or actions that might be taken to build local capacity and contribute to sustainable water management. Students blended scientific knowledge with experiential learning to form their own ILEK. An ERE student who participated in both years wrote of what she gained from the cultural perspective: " . . . I appreciated [learning] about indigenous communities in the Andes . . . a Peruvian graduate student [co-author Yojana Miraya Oscco], who coincidentally was studying at Humboldt, discussed her research [on Indigenous community organizing]. It is super important to be informed about the communities you visit when doing research and something that is neglected far too often in the sciences . . . " This response provides evidence that our students understand the community perspective and leave our program with a more open mind, characteristic of those who practice cultural humility [58].

Bonanza inspired and empowered students, through the explicit integration of scientific and societal needs and leadership training. Research experience, and the satisfaction felt by students who contribute to shared successes, cultivate feelings of belonging, increased social and psychological engagement and ultimately increase interest in pursuing research careers [83–85]. In our case, students were motivated to pursue water resources careers. An ERE student from 2019 wrote, "This trip made me rethink what approach I want to take to grad school (location, concentration), and made me think more about pursuing a career in water resources." Another ERE student from 2019 cited a better understanding of their career goals and increased self-confidence: "This trip solidified my reason for becoming an engineer and broadened my perspective of what an engineer's job is." A geology student from 2019 echoed this increased self-confidence, "This trip broadened my horizons immensely. It made me realize that I can do meaningful work in almost any part of the world while still doing something I enjoy."

Bonanza has empowered students to be leaders in collaborative, community-based and applied research. Four geology and geophysics students presented at the American Geophysical Union Fall Conference in 2018 and 2019. Two geology students graduated from HSU to enter Ph.D. programs, and one geology and 3 engineering students have sought out and found employment in water resources management. The research generated in two seasons of fieldwork is the basis for a master's thesis at HSU [70], and the equipment purchased through this program supported the undergraduate senior theses of two Peruvian students.

## 6. Conclusions

The Bonanza project presents the benefits of including coproduction and application of knowledge within a hydro-social framework. Similar to past models, we explicitly connected upstream and downstream disciplines focused on water resources. What is novel, however, is our focus on the coproduction of knowledge and the application of this knowledge to the benefit of scientists, students, and the local community. Our project successfully addressed two major goals: to contribute directly to sustainable water management within the village of Zurite and to produce an ILEK while training a culturally educated cohort of socio-hydrologists. The resulting ILEK, which incorporates both Allin Kawsay and Western approaches to hydrologic science, has built the local capacity to measure, monitor and manage local water resources, and generated local resilience to natural and socio-political external forces.

Specifically, we explored water resources in the URW, and linked our results to the needs of the community of Zurite. We collaborated with the community to design and construct 1.3 km of irrigation canals, to bring water to the fields of over 100 families, and trained members of the community to monitor local water resources. Based on our study of the URW and the needs of Zurite, we identified key upstream and downstream measures to build ILEK and local resiliency. Upstream, we recommended the conservation of bofedales, to sustain their role as shallow surface reservoirs integral to perennial streamflow. Downstream, we recommend expanding surface water storage, increasing water use efficiency, and examining the impacts of these on water quality.

Our model of knowledge coproduction has had positive and, importantly, lasting educational impacts. We designed and implemented a year-long program that trained 29 students from the U.S. and Perú to be interdisciplinary and community-minded researchers. These students report a change in their life and career outlook, citing increased cultural humility and understanding of local community needs, and have gone into graduate programs and jobs that are focused on water resources. The mutual learning that occurred during different elements of the program created a more complete understanding of the hydrological framework than would have been possible without input from all knowledge-holders.

Based on our work from the Bonanza project, we conclude that combining Western scientific approaches and Allin Kawsay within the ILEK framework can result in impactful changes for water sustainability and can have a lasting impact on all knowledge-holders in the program, including student participants. Our work placed a strong emphasis on local action and defining best practices for sustainable water management and was strengthened by building bridges across different scientific disciplines, and between western scientific approaches and Indigenous and local knowledge, within the community in which we worked. The resulting ILEK provides a framework that can be applied to water development to build local capacity and resiliency and train a culturally educated cohort of socio-hydrologists to work with communities beyond Zurite.

**Author Contributions:** Conceptualization, J.O., K.K. and M.L.; methodology, J.O., K.K., M.L. and Y.M.O.; investigation, J.O., K.K. and M.L; resources, J.O., K.K., M.L. and Y.M.O.; data curation, J.O., K.K., M.L. and Y.M.O.; writing—original draft preparation, J.O., K.K., M.L. and Y.M.O.; writing—review and editing, J.O., K.K., M.L. and Y.M.O.; supervision, J.O.; project administration, J.O.; funding acquisition, J.O., K.K. and M.L. All authors have read and agreed to the published version of the manuscript.

**Funding:** This project was funded by a two-year award from Geoscientists Without Borders (award number: 2017080009) via the Society of Exploration Geophysicists (Founding Supporter Schlumberger). Additional support from North Coast Rotary and the Sponsored Programs Foundation, Humboldt State University. Canal construction costs were supported by GWB, the Municipality of Zurite, the Community of Zurite (San Nicolas de Bari), and the Comisión de Regantes.

**Institutional Review Board Statement:** Not applicable.

**Informed Consent Statement:** Not applicable.

**Data Availability Statement:** No new data were created or analyzed in this study. Data sharing is not applicable to this article.

**Acknowledgments:** This project would not have been possible without the generous support of Tomás Ruiz López and Gladis Quispe. Thank you to the Municipality of Zurite, the Community of San Nicolas de Bari, and the Comisión de Regantes in Zurite. Uriel Ccopa Villena provided essential logistical support. Many undergraduate and graduate students from the United States and Perú contributed to this study. Z. Pearson read an earlier draft and provided many helpful comments that greatly improved the structure and overall quality.

**Conflicts of Interest:** The authors declare no conflict of interest. The funders had no role in the design of the study; in the collection, analyses, or interpretation of data; in the writing of the manuscript, or in the decision to publish the results.

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
