# Peer review of "Interdisciplinary Water Development in the Peruvian Highlands: The Case for Including the Coproduction of Knowledge in Socio-Hydrology"

_hydrology, doi:10.3390/hydrology8030112_

Round 1
Reviewer 1 Report
Dear Authors,
Congratulations for your manuscript. I read it with great interest. Sending an attached doc with a few notes for your consideration.
I have a major point, which is the term 'education' that could be more strongly represented in your article by 'learning' or 'co-learning', since education is quite formal (and traditional way of transmitting knowledge instead of appropriating the knowledge according to the needs).
Besides this point, I think that by the end, and conclusions, you loose some focus away form the communities, and it appears to be more centered only in the students (see my comments) . I strongly suggest that you rewrite some paragraphs in order to make it clear what your approach is, and what kind of alternative 'education' of the students you are envisaging (implying learning from the communities also).
Overall, I found your study very interesting, and relevant, calling attention to the need of considering the local knowledge and the community needs in the learning and training of students.
I wish you all the best in your next steps towards a publication asap.
Reviewer

Author Response
We thank the reviewers for their insightful comments that have helped to improve the manuscript. This document contains our point-by-point response to the reviewer comments. Below, the reviewer comments are in italics and our responses are written in normal font. When we have made changes to the manuscript we provide the line number indicating the location of the change in the marked manuscript.
Reviewer 1:
Dear Authors,
Congratulations for your manuscript. I read it with great interest. Sending an attached doc with a few notes for your consideration.
Thank you for the kind comments. We have addressed your concerns in the revised manuscript.
I have a major point, which is the term 'education' that could be more strongly represented in your article by 'learning' or 'co-learning', since education is quite formal (and traditional way of transmitting knowledge instead of appropriating the knowledge according to the needs).
Thank you for your suggestion. We agree that education is specific to one way knowledge transfer and does not fully represent the structure of the Bonanza program. We’ve adopted language from Sato et al. and our Figure 1 to use knowledge coproduction or coproduction of knowledge when referring to the collaborative knowledge generated by our project.
Besides this point, I think that by the end, and conclusions, you loose some focus away form the communities, and it appears to be more centered only in the students (see my comments) . I strongly suggest that you rewrite some paragraphs in order to make it clear what your approach is, and what kind of alternative 'education' of the students you are envisaging (implying learning from the communities also).
This comment is echoed by the other reviewers and based on these suggestions we have rewritten the results and conclusions to clarify and distinguish knowledge coproduction, and the immediate and long term benefits to students and the community of Zurite. The changes were extensive but the major changes are:
We have rearranged the order of the sections. We moved the upstream and downstream knowledge coproduced to after the description of the educational framework to emphasize that this knowledge was coproduced from this project. We rewrote the conclusions so that it is no longer a summary and now highlights the recommendations that arose as part of the project and to state the conclusions of the project.
Overall, I found your study very interesting, and relevant, calling attention to the need of considering the local knowledge and the community needs in the learning and training of students.
I wish you all the best in your next steps towards a publication asap.
Reviewer
Thank you again for the comments
Reviewer 1 highlighted sections provided in an attached PDF file. Some of the highlighted sections refer to the use of “learning” versus “education”. Other sections were highlighted to indicate sentences that need editing. We have listed the highlighted sections below and have edited them as necessary.
Line 216 to 219: “Non-humans such as animals and plants, but also mountains, valleys, rivers and lakes are integral parts of the community with each one’s individual health serving as a reflection of the health of the community”
This sentence repeated some information that was in the previous sentence and so we have changed it to:
“Furthermore, the health of the mountains, other landforms, plants and animals serves as a reflection of the health of the community.”
Line 332: “with diverse education backgrounds and” was highlighted
We presumed here that the reviewer wanted more specificity as to the diverse educational backgrounds of the students. However, the majors are listed in the previous paragraph and so we have not made changes here.
Line 483-484: “We use qualitative student commentaries to reflect the high-impact practices of embedding our program in academic coursework [62], and increased self-confidence to pursue future research opportunities [63].” was highlighted.
Here we are not certain why the reviewer highlighted this section. We did not collect quantitative data from students but did ask for student feedback, which we connect to previously published works on ‘best practices’ in international research/education.
Line 513-514: “It is super important to be informed about the communities you visit when d” was highlighted.
This highlighted section is a direct quote from a student participant and so we have left it as is.
The following sections were highlighted by the reviewer, but no comment was provided. Without additional information we did not understand what concerns the reviewer had and so have not addressed these highlighted sections.
Line 309: “holistic” was highlighted
Line 518: “empowered students”
Line 531-532: “empowered students to be leaders in collaborative, community-based and applied research.”
Line 545: “c
We thank the reviewers for their insightful comments that have helped to improve the manuscript. This document contains our point-by-point response to the reviewer comments. Throughout this document, the reviewer comments are in blue and our responses are written in black. When we have made changes to the manuscript we provide the line number indicating the location of the change in the marked manuscript.
Reviewer 1:
Dear Authors,
Congratulations for your manuscript. I read it with great interest. Sending an attached doc with a few notes for your consideration.
Thank you for the kind comments. We have addressed your concerns in the revised manuscript.
I have a major point, which is the term 'education' that could be more strongly represented in your article by 'learning' or 'co-learning', since education is quite formal (and traditional way of transmitting knowledge instead of appropriating the knowledge according to the needs).
Thank you for your suggestion. We agree that education is specific to one way knowledge transfer and does not fully represent the structure of the Bonanza program. We’ve adopted language from Sato et al. and our Figure 1 to use knowledge coproduction or coproduction of knowledge when referring to the collaborative knowledge generated by our project.
Besides this point, I think that by the end, and conclusions, you loose some focus away form the communities, and it appears to be more centered only in the students (see my comments) . I strongly suggest that you rewrite some paragraphs in order to make it clear what your approach is, and what kind of alternative 'education' of the students you are envisaging (implying learning from the communities also).
This comment is echoed by the other reviewers and based on these suggestions we have rewritten the results and conclusions to clarify and distinguish knowledge coproduction, and the immediate and long term benefits to students and the community of Zurite. The changes were extensive but the major changes are:
We have rearranged the order of the sections. We moved the upstream and downstream knowledge coproduced to after the description of the educational framework to emphasize that this knowledge was coproduced from this project. We rewrote the conclusions so that it is no longer a summary and now highlights the recommendations that arose as part of the project and to state the conclusions of the project.
Overall, I found your study very interesting, and relevant, calling attention to the need of considering the local knowledge and the community needs in the learning and training of students.
I wish you all the best in your next steps towards a publication asap.
Reviewer
Thank you again for the comments
Reviewer 1 highlighted sections provided in an attached PDF file. Some of the highlighted sections refer to the use of “learning” versus “education”. Other sections were highlighted to indicate sentences that need editing. We have listed the highlighted sections below and have edited them as necessary.
Line 216 to 219: “Non-humans such as animals and plants, but also mountains, valleys, rivers and lakes are integral parts of the community with each one’s individual health serving as a reflection of the health of the community”
This sentence repeated some information that was in the previous sentence and so we have changed it to:
“Furthermore, the health of the mountains, other landforms, plants and animals serves as a reflection of the health of the community.”
Line 332: “with diverse education backgrounds and” was highlighted
We presumed here that the reviewer wanted more specificity as to the diverse educational backgrounds of the students. However, the majors are listed in the previous paragraph and so we have not made changes here.
Line 483-484: “We use qualitative student commentaries to reflect the high-impact practices of embedding our program in academic coursework [62], and increased self-confidence to pursue future research opportunities [63].” was highlighted.
Here we are not certain why the reviewer highlighted this section. We did not collect quantitative data from students but did ask for student feedback, which we connect to previously published works on ‘best practices’ in international research/education.
Line 513-514: “It is super important to be informed about the communities you visit when d” was highlighted.
This highlighted section is a direct quote from a student participant and so we have left it as is.
The following sections were highlighted by the reviewer, but no comment was provided. Without additional information we did not understand what concerns the reviewer had and so have not addressed these highlighted sections.
Line 309: “holistic” was highlighted
Line 518: “empowered students”
Line 531-532: “empowered students to be leaders in collaborative, community-based and applied research.”
Line 545: “community benefits and the training of cohorts of students and community members.”
ommunity benefits and the training of cohorts of students and community members.”
Reviewer 2 Report
In general
This is a good paper. Bringing education into social hydrology is a very good and very important practice. This is killing three birds with one stone. It can be published after carefully check the writing, especially the references.
In details
L67-68: Since you don't have 1.2 etc., it is best to delete 1.1. Or, change it to 3. Method, etc.
L121: Is the “(ANA, 2012)” refer a reference? Please give it.
Figure 7: delete “construction .” it is repeated.
Please give more details of some references such as [10], [17],[22],[23],[25]-[27],[29],[32]-[34],[37],[42], [45], [47], [60], [61], [64], [66], [70], etc. as follows:
L611, References 10: delete Sato T, Kikuchi, N., it is repeated. And please give the reference place, maybe: https://www.chikyu.ac.jp/rihn_e/project/E-05.html。
L718: add:” J. Health Care Poor Underserved. 1998, 9(2):117-25. doi: 10.1353/hpu.2010.0233.”
L728: add “Michigan Journal of Community Service Learning, 1995, pp.112-122”
Author Response
We thank the reviewers for their insightful comments that have helped to improve the manuscript. This document contains our point-by-point response to the reviewer comments. Below, the reviewer comments are in italics and our responses are written in normal font. When we have made changes to the manuscript we provide the line number indicating the location of the change in the marked manuscript.
Reviewer 2:
L67-68: Since you don't have 1.2 etc., it is best to delete 1.1. Or, change it to 3. Method, etc.
We have adjusted the numbering to make it more logical given the sections we have listed. Specifically, we renamed section 1.1 section 2, which necessitated changes to all other section numbering.
L121: Is the “(ANA, 2012)” refer a reference? Please give it.
Yes, that reference refers to a report from the Administación Nacional de Agua - reference is now included.
Figure 7: delete “construction .” it is repeated.
Correction made.
Please give more details of some references such as [10], [17],[22],[23],[25]-[27],[29],[32]-[34],[37],[42], [45], [47], [60], [61], [64], [66], [70], etc. as follows:
L611, References 10: delete Sato T, Kikuchi, N., it is repeated. And please give the reference place, maybe: https://www.chikyu.ac.jp/rihn_e/project/E-05.html。
L718: add:” J. Health Care Poor Underserved. 1998, 9(2):117-25. doi: 10.1353/hpu.2010.0233.”
L728: add “Michigan Journal of Community Service Learning, 1995, pp.112-122”
We have carefully revised the references to make sure that they are all complete and correctly formatted.
Reviewer 3 Report
Dear authors,
I have read this paper with great interest. The topic of water management in rural areas of the developing countries is often of utmost importance for the local communities. Your work has certainly produced some visible advances in water management which will benefit both the local community and the students involved.
But, unfortunately, I see very little scientific outcomes in this paper. I feel that there is a huge gap in scientific methodology, results, discussion and conclusions. The paper appears to be project report rather than research paper. By no means am I discarding your work. You obviously put a lot of effort in this project and you have improved the life of the local people, but there is just not enough scientific "weight" in this paper.
I will next give some specific remarks about the paper.
- Title: I am not sure what "water development" means. How do you develop water? Perhaps "water management development" would be more clear to the readers. I should note that I am not a native English speaker so I might be wrong about this. I'll leave authors to decide.
- lines 119-120 about the moisture and the origins of the westerlies. Your study area is in the tropics and the westerlies form in the mid -latitudes (40-60°) over the oceans far from the Amazon basin. Even if the westerlies were formed over Amazon basin they would transport the moisture away from the Andes not towards them. In the tropics rain season is connected to the migration of the Intertropical Convergence Zone (ITCZ) which "follows" the thermal equator which moves south of geographic equator after the autumn equinox which is why the rain season in your study area is from November to April.
- When you discuss the climate of the area it would be useful to add the climate type according to Köppen-Geiger classification. Your study area is in the high altitude tropics so I'd presume that altitude-modified C-type climate with dry winters is present in the area. Also, a few words about the air temperatures would be nice since the temperature impacts evaporation.
- Line 257: you write about "pre-Inca practices". Is that correct or do you mean pre-European practices.
- Conclusions. This is more summary of a project and not a conclusion of scientific research.
Author Response
We thank the reviewers for their insightful comments that have helped to improve the manuscript. This document contains our point-by-point response to the reviewer comments. Below, the reviewer comments are in italics and our responses are written in normal font. When we have made changes to the manuscript we provide the line number indicating the location of the change in the marked manuscript.
- Title: I am not sure what "water development" means. How do you develop water? Perhaps "water management development" would be more clear to the readers. I should note that I am not a native English speaker so I might be wrong about this. I'll leave authors to decide.
“Water development” refers to water that is developed for human use. It is a term that is commonly used (see for instance the recent United Nations report, “UN World Water Development 2021: Valuing Water”) so we have opted to leave this term in the title.
- lines 119-120 about the moisture and the origins of the westerlies. Your study area is in the tropics and the westerlies form in the mid -latitudes (40-60°) over the oceans far from the Amazon basin. Even if the westerlies were formed over Amazon basin they would transport the moisture away from the Andes not towards them. In the tropics rain season is connected to the migration of the Intertropical Convergence Zone (ITCZ) which "follows" the thermal equator which moves south of geographic equator after the autumn equinox which is why the rain season in your study area is from November to April.
Correct. That is our mistake. Thank you for your explanation. We have changed the language to: “ however, the longitudinal barrier of the Andes forms a barrier between the moist eastern flanks draining to the Amazon Basin and the rain shadow desert along the populous Pacific coast.” .
- When you discuss the climate of the area it would be useful to add the climate type according to Köppen-Geiger classification. Your study area is in the high altitude tropics so I'd presume that altitude-modified C-type climate with dry winters is present in the area. Also, a few words about the air temperatures would be nice since the temperature impacts evaporation.
We have added the Köppen-Geiger climate classification (Cwb).
- Line 257: you write about "pre-Inca practices". Is that correct or do you mean pre-European practices.
Pre-Inca is correct. There are many Andean irrigation customs that predate the Incas. See Ochoa-Tocachi BF, Bardales JD, Antiporta J, et al. Potential contributions of pre-Inca infiltration infrastructure to Andean water security. Nat Sustain. 2019;2(7):584-593. doi:10.1038/s41893-019-0307-1
- Conclusions. This is more summary of a project and not a conclusion of scientific research.
This comment is echoed by the other reviewers and based on these suggestions we have rewritten the results and conclusions to clarify and distinguish knowledge coproduction, and the immediate and long term benefits to students and the community of Zurite. The changes were extensive but the major changes are:
We have rearranged the order of the sections. We moved the upstream and downstream knowledge coproduced to after the description of the educational framework to emphasize that this knowledge was coproduced from this project. We rewrote the conclusions so that it is no longer a summary and now highlights the recommendations that arose as part of the project and to state the conclusions of the project.
Reviewer 4 Report
After carefully reading your paper, I would like to make the following comments:
L80-90: Here you present the upstream and downstream knowledge and what those two include. I see that there is something missing in order to speak about sustainable water management is the impact on the quality and quantity of water resources. I would like to have an insight on this aspect.
L276: What is 0.87 mm? Please rephrase to clarify.
In your study you trained some students in order to raise awareness on water related issues. This practice, although probably the best way to achieve sustainability, has a long term perspective. It would be better to try and contact local stakeholders, involve them in the whole process, provide training and get feedback. Students are an audience that is important but only the local stakeholders can affect the way water resources are managed.
Prevent using words such as chacras or turnos. They might confuse the reader who has to read back to remember what that is.
In your paper it seems that in some parts your writing style is closer to an article that is about to be published in a newspaper rather than a scientific journal. An example is section 2.3.1, where apart from some information related to the water management in the area, there is also criticism about the local policies. These facts have their importance but can distract the reader from the scientific core of the paper.
Finally, in your paper it is not clear what the purpose was and whether it was achieved. Was your goal to train the students and promote interdisciplinary water resources management? Was it to improve infrastructure in the area? If this was your goal, can you quantify the positive impact on the water resources in the area? I understand that it is a socio-hydrology paper but the final results are not very well related to the scientific questions of the paper.
Author Response
We thank the reviewers for their insightful comments that have helped to improve the manuscript. This document contains our point-by-point response to the reviewer comments. Below, the reviewer comments are in italics and our responses are written in normal font. When we have made changes to the manuscript we provide the line number indicating the location of the change in the marked manuscript.
After carefully reading your paper, I would like to make the following comments:
L80-90: Here you present the upstream and downstream knowledge and what those two include. I see that there is something missing in order to speak about sustainable water management is the impact on the quality and quantity of water resources. I would like to have an insight on this aspect.
Since our project primarily focused on sustainable water management for irrigation use, we did not collect measurements to determine the water quality. However, we agree with the reviewer that water quality is an important part of drinking water management and have added a comment in the conclusions that we should examine the water quality in future studies in this region.
L276: What is 0.87 mm? Please rephrase to clarify.
The 0.87 mm is an estimate of the total storage volume applied to all irrigated land. We have removed this value and rewritten the sentence: “The community operates five reservoirs with a total storage capacity of 10,500 m3. This volume is substantially below dry season irrigation demands and necessitates fields being left fallow.”
In your study you trained some students in order to raise awareness on water related issues. This practice, although probably the best way to achieve sustainability, has a long term perspective. It would be better to try and contact local stakeholders, involve them in the whole process, provide training and get feedback. Students are an audience that is important but only the local stakeholders can affect the way water resources are managed.
We absolutely agree that including local stakeholders, in addition to the students, is an important part of addressing long term sustainability within the local community. The Zurite community members were in fact, an integral part of the project. They were included in all stages of the project - from the planning, to the data collection, in the interpretation of results, the design of the canals, etc - and are fundamental to the learning community and the integrated local environmental knowledge plan highlighted in Figure 1. It is unfortunate that this was not entirely clear in the initial manuscript. We think that the lack of clarity may come from our use of the terminology “education” and “local knowledge holders.” In the resubmitted manuscript we now emphasize “knowledge coproduction” and specify that the local knowledge holders are also the local stakeholders, and are a critical component of the project framework.
These changes were made throughout the manuscript, but are primarily in the introduction and when reporting our results. We have added this clarification to section 2 page 2 of the manuscript which now reads:
“In our framework the knowledge actors are the subject matter experts (geologists, geo-physicists, engineers, and hydrologists), the U.S. and Peruvian students, and the local knowledge holders, i.e., the local stakeholders, from the community of Zurite.”
Prevent using words such as chacras or turnos. They might confuse the reader who has to read back to remember what that is.
The term chacras is used twice in the manuscript and the term turnos is used once; all uses are within the same paragraph and the terms are defined immediately following their use. Since the use of these terms is limited we do not think that they will confuse the reader and have left them in the manuscript. If this is acceptable to the reviewer, we’ll leave as is.
In your paper it seems that in some parts your writing style is closer to an article that is about to be published in a newspaper rather than a scientific journal. An example is section 2.3.1, where apart from some information related to the water management in the area, there is also criticism about the local policies. These facts have their importance but can distract the reader from the scientific core of the paper.
Section 2.3.1 (now numbered 3.2.1) was included because understanding the water usage in Zurite requires an understanding of the historical context of water rights within Peru. To make this clear, we have now added language to the beginning of section 3.2.1 to explain the reason for this section. However, as this research is a sociohydrology study, we have not removed these facts as the historical context of water rights and the politics and governance of water in Zurite and in Peru have a strong impact on the water usage in the region.
We have also extensively reorganized the manuscript to clearly present the results: knowledge coproduced, and the immediate and long term benefits to the community and to students. We have edited these sections to make the language more scientific.
Finally, in your paper it is not clear what the purpose was and whether it was achieved. Was your goal to train the students and promote interdisciplinary water resources management? Was it to improve infrastructure in the area? If this was your goal, can you quantify the positive impact on the water resources in the area? I understand that it is a socio-hydrology paper but the final results are not very well related to the scientific questions of the paper.
The goals are stated in the second paragraph of section 2. “Two major goals of the project were sustainable water management within the village of Zurite and a trained cohort of socio-hydrologists. By including knowledge coproduction and the training of an interdisciplinary and multinational cohort of students, the ILEK produced will benefit from the diversity of knowledge actors (in terms of age, subject area expertise, ethnic and cultural background, etc.) who, together will better solve complex environmental problems .” Based on this comment and previous reviewers’ comments, we have rewritten the results and conclusions section to revisit these goals and make this more clear.
Additionally, we have rearranged the order of the sections. We moved the upstream and downstream knowledge coproduced to after the description of the educational framework to emphasize that this knowledge was coproduced from this project. We rewrote the conclusions so that it is no longer a summary and now highlights the recommendations that arose as part of the project and to state the conclusions of the project.
Round 2
Reviewer 3 Report
Ths version is far more suitable for publishing than the original one. And more in line with the classical scinetific paper's structure nad methodology.
Reviewer 4 Report
I have had the time to go through the updated version of your paper. I see that there are improvements, especially in the conclusion part, making the results more clear to the reader. I still find some odd things in this paper, e.g. Allin Kawsay is mentioned but no specific details on what are the practices used under this philosophy, or why there is a specific mention to Dr Oshun's volunteer work (L497). I think my comments on the style are not addressed but since the Editors find that this suits the journal it is beyond the reviewer's role.